

# Hybrid mRMR and multi-objective particle swarm feature selection methods and application to metabolomics of traditional Chinese medicine

Mengting Zhang[1], Jianqiang Du[1,2], Bin Nie[1,2], Jigen Luo[1,2], Ming Liu[1] and Yang Yuan[1]

[1] School of Computer Science, Jiangxi University of Chinese Medicine, Nanchang, China
[2] Key Laboratory of Artificial Intelligence in Chinese Medicine, Jiangxi University of Chinese Medicine, Nanchang, China

## ABSTRACT

Metabolomics data has high-dimensional features and a small sample size, which is typical of high-dimensional small sample (HDSS) data. Too high a dimensionality leads to the curse of dimensionality, and too small a sample size tends to trigger overfitting, which poses a challenge to deeper mining in metabolomics. Feature selection is a valuable technique for effectively handling the challenges HDSS data poses. For the feature selection problem of HDSS data in metabolomics, a hybrid Max-Relevance and Min-Redundancy (mRMR) and multi-objective particle swarm feature selection method (MCMOPSO) is proposed. Experimental results using metabolomics data and various University of California, Irvine (UCI) public datasets demonstrate the effectiveness of MCMOPSO in selecting feature subsets with a limited number of high-quality features. MCMOPSO achieves this by efficiently eliminating irrelevant and redundant features, showcasing its efficacy. Therefore, MCMOPSO is a powerful approach for selecting features from high-dimensional metabolomics data with limited sample sizes.

## INTRODUCTION

Metabolomics is a relatively new area of study, and the tools for best practices and data analysis are still evolving, making metabolomics data analysis a critical task in biomedical research (*Cambiaghi, Ferrario & Masseroli, 2017*). Metabolomics data is often characterized by high feature dimensionality but with limited samples, making it a typical high-dimensional small sample (HDSS) problem. Achieving specific tasks in metabolomics, such as predicting biomarkers (*Grissa et al., 2016*) or identifying effective compounds overfitting, can be quite challenging. At this point, it is necessary to propose advanced solutions with the help of data mining methods. Indeed, feature selection is not only a valid method for addressing HDSS problems but also serves as an essential dimensionality reduction technique. However, some feature methods cannot eliminate

Corresponding author
Jianqiang Du,
jianqiang_du@163.com

redundant features, lack stability, lead to unstable variations in the selected feature subsets, and are unsuitable for HDSS data.

HDSS learning has been essential in statistical machine learning research for many years. HDSS learning has been crucial in statistical machine learning research. Feature selection is a stand-alone approach for processing high-dimensional data and has been broadly utilized in multiple fields, like genomics (*Afshar & Usefi, 2020*), biological data (*Mafarja et al., 2023*), and credit risk assessment (*Yu, Yu & Yu, 2021*). Building upon the various approaches of combining feature selection with learning algorithms, feature selection methods can be classified into four categories: filtered, wrapped, embedded, and integrated. Filtered feature selection does not rely on a learning algorithm; some evaluation criteria rank all features, and all features are ranked by some evaluation criterion rank all features and a portion of the features are output after the ranking. The evaluation methods based on feature ranking are the maximal information coefficient (MIC) (*Sun et al., 2018*), Pearson correlation coefficient, Spearman correlation coefficient, mutual information, and Max-Relevance and Min-Redundancy (mRMR) (*Gu et al., 2022*) *etc.* Wrapper-based feature selection methods involve integrating with a learning algorithm by encapsulating it as a black box. The quality of selected features is evaluated based on the prediction accuracy of the learning algorithm on the feature subset. Learning algorithms include support vector machines in classification problems (*Benítez-Peña et al., 2019*), partial least squares regression (PLS), Lasso (*Li, Lai & Cui, 2021*), and heuristic search algorithms in regression problems. Embedded feature selection is integrated within the learning algorithm and compromises the first two approaches, including ID3 (*Zhu & Zhong, 2010*), CART (*Dong et al., 2021*), *etc.* The integrated feature selection method incorporates multiple feature selection methods, drawing on the concept of integrated learning. This approach can combine the strengths of each algorithm, making it applicable not only to HDSS data but also to improving algorithm stability. However, it may come with a trade-off in terms of time performance. For example, *Li et al. (2022)* integrated the use of MIC and approximate Markov blankets and L1 regular terms (DA2MBL1), which can effectively solve the HDSS problem.

Feature selection contains two objectives: minimizing the size of the feature subset and maximizing the model accuracy. Many methods focus on pursuing the precision of the model while ignoring the size of feature subsets. To balance the two objectives, researchers treat it as a MOOP. Currently, multi-objective feature selection has received much attention. For example, an enhanced multi-objective immune algorithm (MOIA) is proposed in the literature (*Wei et al., 2020*) for feature selection in intrusion detection. The literature uses the multi-objective evolutionary Evolutionary Non-Dominated radial-based algorithm (ENORA) as a wrapper approach for the search strategy to solve the online sales forecasting problem (*Jiménez et al., 2017*). A discrete sine cosine algorithm (SCA)-based multi-target feature selection (MOSCA_FS) method for hyperspectral images was proposed in the literature (*Wan et al., 2020*) *etc.*

In HDSS problems, using filtered feature selection or multi-objective feature selection methods alone cannot effectively remove irrelevant and redundant features. Researchers have done much research to find a solution to the HDSS feature selection problem. *Zhang*

*et al. (2022)* proposed a data augmentation and hybrid feature selection method based on Wasserstein generative adversarial networks (WGAN) for credit risk assessment in HDSS scenarios. In their study, *You, Yang & Ji (2014)* were concerned with feature selection in HDSS scenarios. They proposed a comprehensive analytical framework designed for feature selection in such domains. The framework encompasses selection strategies, including single-feature and multi-feature ranking, and evaluation criteria, such as feature subset uniformity and compact size. Additionally, they introduced a feature selection method based on partial least squares tailored specifically for HDSS data. Notably, they derived and presented two theorems to enhance the understanding and effectiveness of their proposed methodology. *Zhang & Cao (2019)* adopted a filter feature selection algorithm established on redundancy removal (FSBRR) to classify high-dimensional biomedical data. These methods suit the HDSS problems and effectively remove irrelevant features to improve the model's accuracy. However, some redundant features remain, and the resulting feature subset is not streamlined enough. Therefore, it is necessary to study new HDSS feature selection methods to effectively eliminate irrelevant and redundant features and filter a set of streamlined and stable feature subsets. This will ultimately enhance model precision and stability.

This study presents a hybrid feature selection method called multi-objective particle swarm feature selection method (MCMOPSO) to maximize the removal of irrelevant and redundant features. By combining the strengths of mRMR and MOPSO, this approach strives to identify an optimal subset of features. MCMOPSO is segmented into two stages. The first phase uses a filtered feature selection method—mRMR, to rank feature subsets and a wrapper approach to adaptively remove irrelevant and partially redundant features, greatly reducing the size of feature subsets. In the second stage, a MOPSO feature selection method building on dynamic acceleration factors and nonlinear decreasing inertia weights is used to filter the feature subset, which not only removes the redundant features based on the first stage but also enhances the precision of the model. The experiments show that the MCMOPSO method can effectively solve the feature selection problem of HDSS and provide a powerful help for metabolomics work. The specific contributions of this article are outlined as follows:

(1) For the problem of HDSS in feature selection, a new hybrid feature selection method, MCMOPSO, is proposed. The process can effectively remove redundant and irrelevant features and can be taken as a fundamental framework for feature selection of HDSS data.

(2) To solve the problem of local optima in the multi-objective particle swarm optimization algorithm in the second stage of MCMOPSO, a MOPSO algorithm based on dynamic linear adjustment of acceleration factors and nonlinear decreasing weight coefficients (CMOPSO) is proposed to obtain better diversity at the early stage to let the particles get rid of local optima, and to enhance the performance of MOPSO, the convergence speed of particles can be accelerated in the later stages.

(3) To verify that MCMOPSO can validly remove irrelevant and redundant features and enhance the model's accuracy, two metabolomics datasets and 10 HDSS datasets were used for experiments. The validity of CMOPSO was verified using three conventional data sets.

In this article, the work related to this study will be introduced in "Materials and Methods", the new model proposed in this article will be specified in "Experiment Design and Analysis", and three conventional datasets, two metabolomics data, and 10 individual conventional HDSS datasets will be used in "Discussion". The new model is subjected to experimental analysis from various perspectives. Three algorithms are compared in the first stage, while two in the second stage are compared. Furthermore, MCMOPSO is compared to two other two-stage feature selection methods to validate the feasibility and effectiveness of the new model further. The fifth section provides a comprehensive summary of the entire article.

# MATERIALS AND METHODS

## Basic concepts

### mRMR

mRMR (*Peng, Long & Ding, 2005*) is to find the set of features in the original set of features that have the highest relevance to the final output result (Max-Relevance) and the set of features with the minor correlation between features (Min-Redundancy). Assuming that x and y are two random variables, the mutual information is defined as:

$$I(x;y) = \iint p(x,y) \log \frac{p(x,y)}{p(x)p(y)} dxdy. \tag{1}$$

The maximum class correlation $\max(D(S,C))$ for the target class C and a subset of features with S features is defined by the average of the correlations between the selected features $x_i(i = 1, 2, \ldots, S)$ and the target class C:

$$\max(D(S,C))\ D = \frac{1}{s} \sum_{x_i \in S} I(x_i, C). \tag{2}$$

The following equation can evaluate the redundancy of S:

$$\min R(s) = \frac{1}{|S^2|} \sum_{(x_i, x_j) \in S^2} I(x_i, x_j). \tag{3}$$

The mRMR ranks features by simultaneously maximizing relevance and minimizing redundancy and is expressed in the form of the following equation:

$$\text{mRMR} = \max \left[ D = \frac{1}{s} \sum_{x_i \in S} I(x_i, C) - R = \frac{1}{|S^2|} \sum_{(x_i, x_j) \in S^2} I(x_i, x_j) \right]. \tag{4}$$

### Multi-objective optimization

Various complicated tasks need to optimize two or more objectives simultaneously, and the objective functions conflict with each other and cannot explicitly balance them; that is,

the optimization of one objective may cause the decay of another objective and cannot make each objective function optimal. The challenge of simultaneously optimizing conflicting objectives within a defined range is called a multi-objective optimization problem (MOOP) (*Lücken Von, Brizuela & Barán, 2019*). The goal is to find solutions to achieve the best possible trade-off among the competing objectives. When solving practical problems with multi-objectives, feature selection will be considered a maximization or minimization problem. If the number of objectives is n, a standard multi-objective optimization mathematical model should be denoted as shown in Eq. (1). Multi-objective optimization methods aim to obtain a set of non-dominated optimal solutions or a subset of characteristics.

$$\min(\max)F(x) = \{f_1(x), f_2(x)...f_n(x)\}$$
$$s.t.x \in \Omega$$

(5)

The Pareto dominance method (*Newman, 2005*) is a standard method for solving MOOP. It is defined by examining two decision vectors $a, b \in X$, $a$ Pareto dominance $b$, noted as $a > b$, when and only when the following equation, which can also be called $a$ dominates $b$. If no other decision variable can dominate it, then the decision variable is termed a non-dominated solution.

$$\{\forall i \in \{1, 2, ..., n\} f_i(a) \leq f_i(b)\} \wedge \{\exists j \in \{1, 2, ..., n\} f_j(a) < f_j(b)\}. \tag{6}$$

### Particle swarm optimization

Particle swarm optimization (PSO), as described in the work previously published by *Figueiredo, Ludermir & Bastos-Filho (2016)*, is a type of swarm intelligence algorithm inspired by the foraging behavior of bird flocks. A weightless particle simulates birds within a flock, possessing two properties: velocity (V) and position (X). The velocity attribute of the particle corresponds to the magnitude or speed at which it moves, while the position attribute indicates the direction or orientation of its movement. Each particle conducts an individual search within the search space to find its personal best solution, which is known as the $P_{best}$. This $P_{best}$ is then shared among all particles in the swarm. Together, they determine the current global best ($G_{best}$) solution, representing the overall best solution found by the entire swarm. Each particle adjusts its velocity and position based on its $P_{best}$ and the current $G_{best}$ solution shared among the particles. The position of the particle is $X_i = (x_{i1}, x_{i2}, ..., x_{iD})$, the velocity is $v_i = (v_{i1}, v_{i2}, ..., v_{iD})$, the individual extremum is $P_i = (p_{i1}, p_{i2}, ..., p_{iD})$ and the global optimal solution is $P_g = (p_{g1}, p_{g2}, ..., p_{gD})$. Equations give the updated velocity and position formulas:

$$V_{id}(t+1) = w * V_{id}(t) + c_1 * r_1 * (p_{id}(t) - x_{id}(t)) + c_2 * r_2 * (p_{gd}(t) - x_{id}(t)) \tag{7}$$

$$x_{id}(t+1) = x_{id}(t) + V_{id}(t+1) \quad i = 1, 2, ..., N, d = 1, 2, ..., D \tag{8}$$

where t denotes the number of iterations and N is the size of the population.

## The proposed method

MCMOPSO encapsulates Filter and Wrapper feature selection methods and is divided into two main phases. In the first phase, a combination of Filter and Wrapper removes irrelevant and partially eliminates redundant features. This adaptive approach aims to identify the best-performing candidate feature subset; the second phase uses a heuristic search algorithm approach to remove redundant features and find a refined dataset using a MOPSO algorithm. The construction flow of the algorithm is shown in Fig. 1.

In the first phase of MCMOPSO, mRMR calculates the correlation between each feature $f_i$ and the target feature $Y$ and the redundancy between $f_i$ and $f_j$. These correlations and redundancies are then sorted based on a combined measure of maximum relevance and minimum redundancy. Subsequently, the forward search strategy (SFS) is employed, gradually adding a predetermined number of features. After adding each set of features, PLS regression is employed to model the feature subset, and 10-fold cross-validation is used to evaluate the performance of PLS and avoid overfitting or under-fitting issues. Finally, the candidate feature subset is selected based on the minimum RMSE value $RMSE_{best}$ corresponding to the PLS regression.

In the second phase of MCMOPSO, a heuristic search strategy is employed to remove redundancy using a MOPSO algorithm. This strategy aims to balance the size of the feature subset and the RMSE of PLS. As a result, a set of non-dominant solutions (*Paretopops*) is obtained, achieving the objective of eliminating redundant features and obtaining the optimal feature subset. The same 10-fold cross-validation is used, where the dataset is divided into 10 parts, and each time nine parts of data are used for training and one part of data is used for testing. This process is executed 10 times repeatedly, each time choosing a different test set and training set, and ultimately obtaining the average of the RMSEs, which is used to measure the performance of the model.

### Fitness function

Currently, the PSO algorithm is mainly applied to continuous optimization problems, and the MOOP has been studied less. Feature selection has two conflicting objectives, feature subset minimization, and performance maximization (*Zhang et al., 2023*), and is a typical MOOP.

In this study, the number of feature subsets and the root mean square error (RMSE) of the evaluation index of the regression task are chosen as the objective functions of the MOPSO, where the number of feature subsets is denoted as $f_1$. The RMSE is denoted as $f_2$, as shown in Eqs. (9) and (10).

$$f_1 = num \tag{9}$$

$$f_2 = \sqrt{\frac{1}{n}\sum_{i=1}^{n}\left(Y - \hat{Y}\right)^2} \tag{10}$$

where, *num* denotes the number of selected features, $Y$ is the predicted value, and $\hat{Y}$ is the actual value.

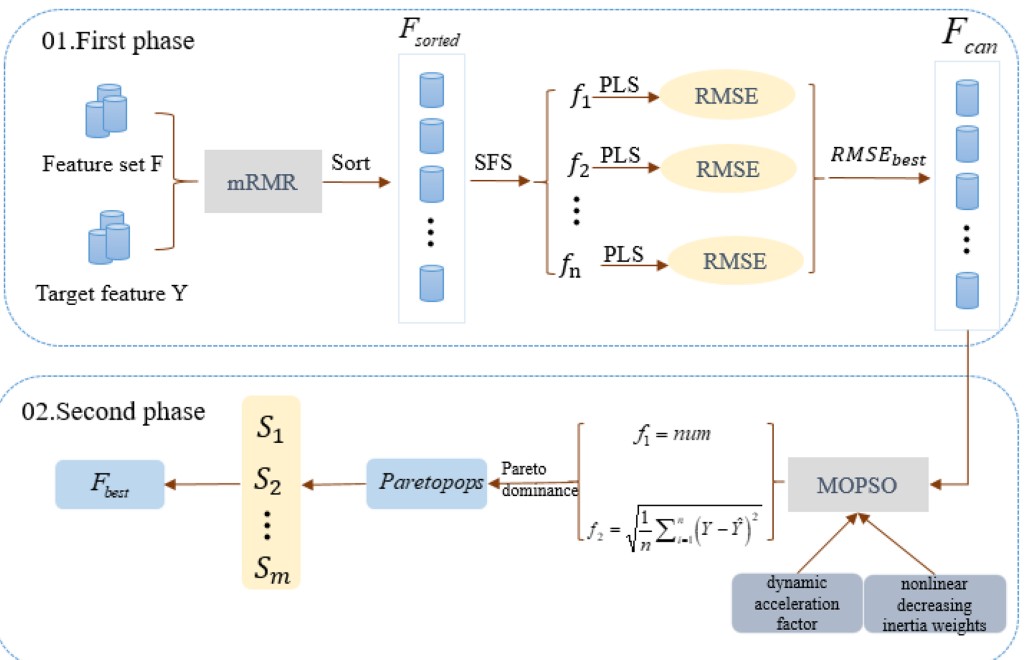

**Figure 1 Algorithm flowchart for the two-stage algorithm MCMOPSO.** This chart depicts the flowchart of the MCMOPSO two-stage algorithm.

### Multi-objective particle swarm optimization based on dynamic acceleration factor and nonlinear decreasing inertia weights (CMOPSO)

To address the problem of the MOPSO algorithm quickly falling into the local optimum, this article improves the convergence of MOPSO by using the dynamic acceleration factor and nonlinear decreasing inertia weights to make the locally optimal particles jump out. The specific flow of the algorithm is outlined in Algorithm 1.

The equations for the nonlinear decreasing inertia factor (w) value and the dynamic acceleration factor are shown in Eqs. (11) and (12), where r1 and r2 refer to two random values within [0, 1], t denotes the number of iterations, and T denotes the total number of iterations. The change in the magnitude of the nonlinearly decreasing inertia factor value (w) with an increasing number of iterations is shown in Fig. 2. In the literature (*Feng, Chen & Guo, 2006*), the range of the acceleration factor is obtained: when $c_1 = 2.75 \sim 1.25$ and $c_2 = 0.5 \sim 2.25$, the effect is better, so the parameters $c_{1f} = 2.75$, $c_{1i} = 1.25$, $c_{2f} = 0.5$ and $c_{2i} = 2.25$ are set. Then, the size changes with the number of iterations, as shown in Fig. 3, with the number of iterations increasing linearly and decreasing, with the number of iterations increasing and growing linearly.

$$w = (w_{\max} - w_{\min}) * (t/T - 1)^2 + w_{\min} \tag{11}$$

$$c_1 = c_{1f} + (c_{1i} - c_{1f}) * (t/T)$$
$$c_2 = c_{2f} + (c_{2f} - c_{2i}) * (t/T) \tag{12}$$

---

| Algorithm 1 Multi-objective particle swarm based on dynamic acceleration factor and nonlinear decreasing inertia weights, CMOPSO. |
| --- |
| Input: Iteration number(T), population size(nPop), Maximum size of archive set(nAr), Size of particles(nChr) |
| Output: Pareto solution set *Paretopops* |
| Step 1: Initialize the population, including initializing the speed and position of $G_{best}$, $P_{best}$ and particles in the population. Set the correlation coefficient |
| Step 2: Get an Archive based on the Pareto dominance principle |
| Step 3: The velocity and position of particles in the population are updated by Formulas (7) and (8) |
| Step 4: Update $P_{best}$ and $G_{best}$ according to fitness function |
| Step 5: Calculate the particle crowding distance and sort the particles according to the crowding distance |
| Step 6: Update Archive |
| Step 7: While the termination criterion is not fulfilled, do Step3 |
|       return *Paretopops* in Archive |

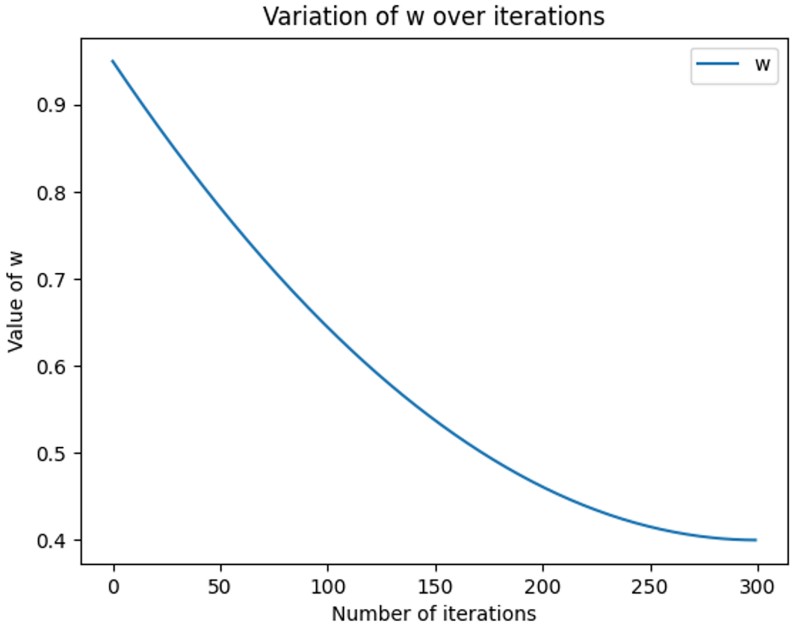

**Figure 2 Graph of nonlinear decreasing inertia weight (w) values with increasing number of iterations.** The graph shows how w varies with the number of iterations in each iteration, based on the given maximum and minimum values, calculated using the quadratic function. In the graph, the x-axis represents the number of iterations, ranging from 0 to nIter. the y-axis represents the value of w.

### Hybrid mRMR and multi-objective particle swarm feature selection methods

To address the feature selection problem in HDSS settings, this article introduces a two-stage method called MCMOPSO. During the first phase of MCMOPSO, the mRMR algorithm measures the correlation between features and the target feature and the redundancy among features. By prioritizing maximum relevance and minimum redundancy, the features are sorted. The Wrapper strategy is then employed to eliminate

---

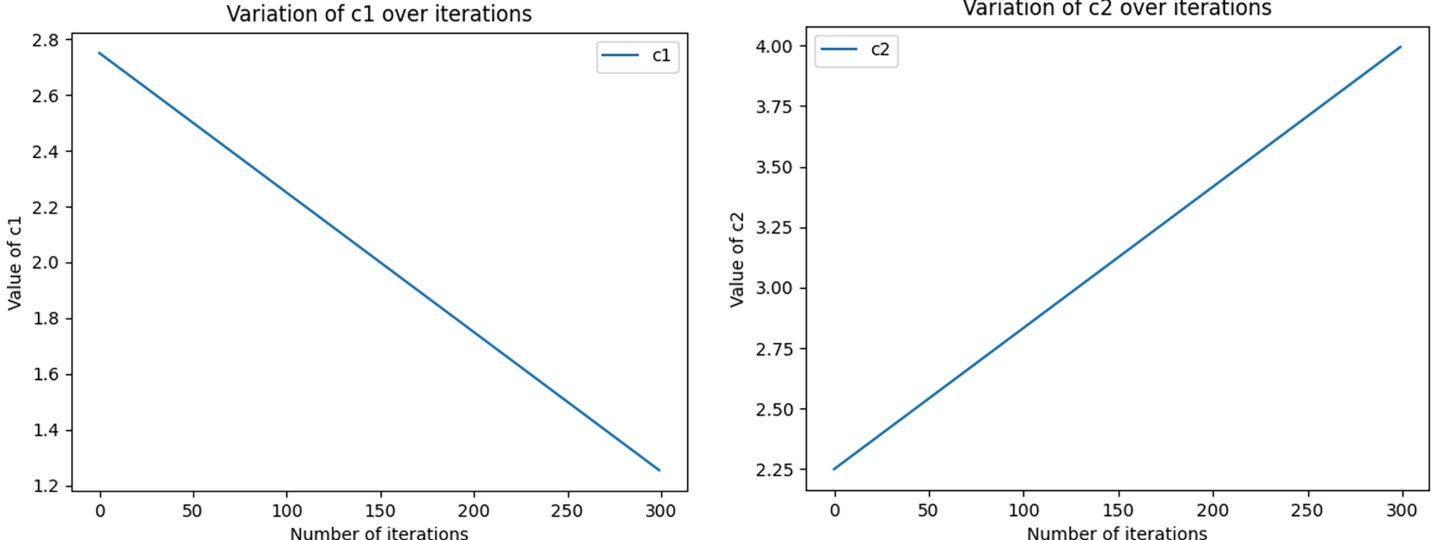

**Figure 3** **The size of the acceleration factors c1 and c2 varies dynamically with the number of iterations.** Initial values (c1i, c1f, c2i, and c2f) as well as the total number of iterations (nIter) are used to compute the values of c1 and c2 in each iteration, which are derived by means of linear interpolation. In the graph, the x-axis represents the number of iterations ranging from 0 to 100. y-axis represents the magnitude of c1 and c2. As the number of iterations increases, the size of c1 gradually approaches the final value, while the size of c2 gradually approaches the initial value.

irrelevant and partially redundant features, enabling the adaptive discovery of the best-performing candidate feature subset. In Phase 2, the remaining redundant features are removed from the MOPSO based on dynamic acceleration factors and nonlinear decreasing inertia weights, and a subset of features with a small number and high accuracy is selected.

Assuming that there are $m$ features, $n$ samples, a subset of features after data pre-processing $F = (f_1, f_2, \ldots, f_m)$, and sample features after pre-processing $Y = (y_1, y_2, \ldots, y_n)^T$, the specific construction process of the MCMOPSO model is as follows:

Phase 1: Filter coupled with Wrapper to eliminate irrelevant and partially redundant features

Step1. mRMR calculates correlation and redundancy: the mRMR scores between each feature $f_i$ and the target features $Y$ and $f_i$ are calculated to obtain the score sequence $mRMR_{list} = (f_1 : mRMR(f_1, Y), f_2 : mRMR(f_2, Y), \ldots, f_m : mRMR(f_m, Y))$, and the score sequence $mRMR_{list}$ is sorted in descending order.

Step2. Determine the subset of candidate features: To address the challenge of determining the number of features and threshold values in the Filter method; this article incorporates the Wrapper approach to determine the optimal number of features to retain adaptively. In this method, a certain number of features are added by a forward search strategy. Then, PLS is used to model the regression of feature subsets, and finally the one with the smallest RMSE value is the candidate feature subset $F_{can}$.

Phase 2: Use a heuristic search algorithm to remove redundancy again

Step3. Eliminate redundant features: The proposed MOPSO optimization algorithm, which incorporates dynamic acceleration factors and non-linearly decreasing inertia weights, is employed to remove redundant features from the candidate feature subset ($F_{can}$). This process results in the identification of the optimal feature subset ($F_{best}$).

Step4. Model evaluation: The effectiveness and timeliness of the model are evaluated by assessing the performance of the optimal feature subset.

The complete MCMOPSO algorithm is shown in Algorithm 2:

## EXPERIMENT DESIGN AND ANALYSIS

All experiments were run on the same personal computer, and the specific configuration information of the computer is shown in Table 1.

### Metabolomics data and UCI data description

To test the validity of CMOPSO, this article first uses three conventional data sets, Residential Building Data Set (abbreviated as RBuild) and Student Performance Data Sets (with math scores and Portuguese scores, abbreviated as SPMath and SPPortuguese, respectively) from the UCI data set. The data information is shown in Table 2 after the missing value processing. Conventional datasets have been widely validated and applied in different domains, and their data characteristics differ significantly from those of metabolomics data. Therefore, the performance of CMOPSO on these datasets can provide a more comprehensive evaluation of the algorithm rather than being limited to domain-specific data. In addition, the selection of traditional datasets facilitates comparisons with pre-existing algorithms to assess the strengths and weaknesses of the CMOPSO algorithm relative to existing methods. It ensures that the algorithms perform as expected on standard datasets and are general and generalizable.

Second, metabolomics data from HDSS were used to test the proposed MCMOPSO algorithm. This article used experimental data on the material basis of ginseng injection for the treatment of cardiogenic shock from the Center for the Development of Differentiation of Basic Theories of Traditional Chinese Medicine at the Jiangxi University of Chinese Medicine. The metabolomics data information for the HDSS is described in Table 3. Experimental data were obtained by waters Hclass high-performance liquid chromatography and synapt G2-si mass spectrometer. LC-MS-QTOF in negative ion mode was used for experimental data acquisition. The aqueous phase A in the chromatographic conditions was ultrapure water (containing 0.1% formic acid), and the organic phase B was acetonitrile. The column temperature was 40 °C, the sample chamber temperature was 10 °C, the flow rate was 0.4 ml/min, and the injection volume was 1 ul. In the mass spectrometry conditions, the ionization source temperature was set at 100 °C, the cone well gas was nitrogen with a flow rate of 50 L/h, and the desolventizing gas was nitrogen with a temperature of 400 °C and a flow rate of 800 L/h. In the positive ionization mode, the capillary voltage was 3.0 kV, the cone well voltage was 40 V, and the extracting cone well voltage was 80 V. The chromatographic data were collected by LC-MS-QTOF. The extraction cone pore voltage was 80 V, the collection time range was 0–25 min, and

---

**Algorithm 2** Feature Selection of hybrid mRMR and MOPSO, MCMOPSO.

Input: Feature set $F = (f_1, f_2, \ldots, f_m)$, target feature $Y = (y_1, y_2, \ldots, y_n)^T$

Output: optimal feature subset $F_{best}$

Phase 1: filtering irrelevant and partial redundancy features and selecting feature subset $F_{can}$

Step 1: Initialize $mRMR_{list} = \{\}, F_{can} = \{\}$

Step 2: for $f_i$ in $F$:

Step 3:     Calculate the relevance and redundancy $mRMR(f_i, Y)$ of $f_i$ and $Y$

Step 4.     $mRMR_{list} = mRMR_{list} + \{f_i : mRMR(f_i, Y)\}$

Step 5: end for

Step 6: $F_{sorted} = SortByDesc(mRMR_{list})$

Step 7: $F_{can}$ adds the features of $F_{sorted}$

Step 8: PLS was used to conduct regression modeling for $F_{can}$ and to save the predicted performance of the corresponding number of feature subsets

Step 9: Combine the forward search strategy to add one feature count at a time and perform step 7

Phase 2: removing redundant features to get the optimal feature subset $F_{best}$

Step 10: Initialize $F_{best} = \{\}$

Step 11: Use Algorithm 1 to get $Paretopops$

Step 12:   $F_{best} = Paretopops$

Step 13. return $F_{best}$

---

**Table 1 The computer configuration information.**

| Computer configuration | Information |
|---|---|
| CPU | Inter(R)-Core(TM)-i7-8700 |
| Frequency | 3.2 GHz-CPU |
| RAM | 16.0 GB |
| Operating system | Windows-10 (64 Bit) |
| Language | Python 3.10.9 |

**Table 2 Description of the general dataset information used in the study.**

| Dataset | Sample (n) | Feature (n) | Target feature (n) |
|---|---|---|---|
| RBuild | 372 | 103 | 2 |
| SPMath | 395 | 30 | 3 |
| SPPortuguese | 649 | 30 | 3 |

Note:
Sample column represents the number of samples in the dataset, feature column is the number of features in the dataset, and target feature is the number of target features.

the mass number range was 50–1,200 Da. In the negative ion mode, the capillary voltage was 2.5 kV, the cone pore voltage was 40 V, and the compensation voltage was 80 V.

The model group, the blank group, and the dosing group were set up respectively, in which seven doses of ginseng injection were given to the shock model rats at 0.1, 0.33, 1.0,

**Table 3 Description of the metabolomics HDSS dataset information used in the study.**

| Dataset | Sample (n) | Feature (n) | Target feature (n) |
|---|---|---|---|
| Endo | 54 | 10,283 | 4 |
| Exo | 42 | 798 | 4 |

3.3, 10, 15, and 20 (unit: ml-kg-1), with six samples of each dose. After 60 min of drug administration, drug efficacy indexes were collected: hemoglobin flow rate ($\mu m/s$), vascular tube diameter ($\mu m$), leukocyte adhesion number (pcs), and vascular permeability, which were used as four dependent variables, namely the four target features. Information on detected substances is characterized by a general division into two parts: a part for substances contained in the experimental rats themselves, called endogenous substance (Endo), which requires the use of blank and model groups for comparison. The other part is information about the substances contained in the ginseng injection, called exogenous substance (Exo), and only seven different doses of samples are needed for comparison.

Finally, to confirm the generalizability of the MCMOPSO algorithm, data from 10 small high-dimensional samples are used to verify the effectiveness of MCMOPSO, which come from the BlogTeFeedback Data Set on the UCI dataset (abbreviated as BlogTe, the BlogTeFeedback Data Set has 50 test sets named by date, and the first 10 are selected here). Table 4 presents the information description of each dataset after undergoing data preprocessing.

## Comparison of mRMR with other filtering algorithms

When a feature is selected, two problems may be faced: how relevant the features are to the category prediction and how redundant the features are to each other. Applying heuristic algorithms directly to HDSS data can pose challenges, like the risk of overfitting and high computational complexity, which may hinder the effective elimination of irrelevant and redundant features. Therefore, we adopt combined filter and wrapper feature selection methods to eliminate irrelevant and partially redundant features. This section uses four filtering methods, MIC, Pearson, spearman, and mRMR, to rank the features and combine them with the Wrapper strategy to eliminate irrelevant and partially redundant features and find the best-performing candidate feature subset adaptively.

The results are shown in Table 5, with black bolded values representing the best results. The endogenous substances in the experimental data were selected from less than 300 ions out of the original 10,283 ions, and the exogenous substances were selected from less than 200 ions out of the original 798 ions, except for y3. Among them, the mRMR algorithm has the lowest RMSE in all seven datasets, but the RMSE in Exo-y2 is higher than that of the MIC and Spearman algorithms. Therefore, from the above experimental results, it can be seen that the mRMR algorithm performs better in dealing with HDSS data by selecting fewer ions with high quality, $i.e.$, small RMSE. In addition, this article also compares the R-squared and mean absolute error (MAE) results in File S1 (Table S3), and the results, six

**Table 4 Description of the information used in the study regarding the HDSS dataset on UCI.**

| Dataset | Sample (n) | Feature (n) | Target feature (n) |
|---------|-----------|-------------|--------------------|
| BlogTe1 | 115 | 280 | 1 |
| BlogTe2 | 133 | 280 | 1 |
| BlogTe3 | 116 | 280 | 1 |
| BlogTe4 | 103 | 280 | 1 |
| BlogTe5 | 92 | 280 | 1 |
| BlogTe6 | 83 | 280 | 1 |
| BlogTe7 | 135 | 280 | 1 |
| BlogTe8 | 155 | 280 | 1 |
| BlogTe9 | 181 | 280 | 1 |
| BlogTe10 | 141 | 280 | 1 |

**Table 5 Comparison of mRMR with other filtering algorithms results obtained at one stage in metabolomics data.**

| Dataset | Feature number (n) | | | | | RMSE | | | | |
|---------|---------|------|---------|----------|------|----------|----------|----------|----------|----------|
| | Full set | MIC | Pearson | Spearman | mRMR | Full set | MIC | Pearson | Spearman | mRMR |
| Endo-y1 | 10,283 | 6,170 | 1,234 | 2,982 | **102** | 741.7310 | 698.2950 | 660.3764 | 664.6072 | **454.1142** |
| Endo-y2 | 10,283 | 4,216 | 1,028 | 1,440 | **102** | 29.4566 | 25.9930 | 27.1033 | 24.0859 | **14.4735** |
| Endo-y3 | 10,283 | 4,216 | 2,262 | 2,674 | **102** | 5.1288 | 4.9224 | 4.8017 | 5.0549 | **3.4551** |
| Endo-y4 | 10,283 | 1,028 | 1,028 | 1,542 | **205** | 10.9573 | 9.5462 | 9.0001 | 9.3926 | **8.0797** |
| Exo-y1 | 798 | 471 | 80 | 551 | **79** | 352.0239 | 253.2525 | 226.6377 | 271.5681 | **242.6975** |
| Exo-y2 | 798 | 327 | 782 | **96** | 791 | 19.0489 | 17.2848 | 20.2707 | **15.3986** | 19.0935 |
| Exo-y3 | 798 | 782 | 766 | 782 | **191** | 3.5128 | 3.5575 | 3.5317 | 3.5989 | **2.7900** |
| Exo-y4 | 798 | 112 | 782 | 782 | **79** | 3.6348 | 3.3443 | 3.9640 | 3.6721 | **3.0507** |

**Note:**
Values in bold means best results.

data in the metabolomics dataset, have better R-squared and MAE than the other algorithms. Therefore, the mRMR algorithm outperforms the other compared algorithms on the metabolomics dataset.

Since mRMR can rank features based on their relevance to the sample and redundancy between features, the feature subset size is much reduced compared to the complete set. From Table 6, it can be seen that on the BlogTe dataset, mRMR has the least features on BlogTe1-2 and BlogTe6 and the second-best performance on BlogTe5 and BlogTe8-10. The results (Table S4 in File S1) in R-squared and MAE are also not as good as those of other algorithms. Although mRMR does not perform optimally on the BlogTe dataset, on the whole, it faces higher-dimensional and more complex metabolomics data, the size of the filtered feature subset is smaller, and the performance of the filtered feature subset is better. mRMR is, therefore, more suitable for high-dimensional data compared than the other three feature ranking methods, *i.e.*, a streamlined feature set can be selected using the mRMR feature selection method.

**Table 6  Comparison of mRMR with other filtering algorithms Results obtained at one stage in UCI conventional HDSS data.**

| Dataset | Feature number (n) | | | | | RMSE | | | | |
|---|---|---|---|---|---|---|---|---|---|---|
| | Full set | MIC | Pearson | Spearman | mRMR | Full set | MIC | Pearson | Spearman | mRMR |
| BlogTe1 | 280 | 31 | **28** | **28** | **28** | 36.3218 | **26.6553** | 29.2033 | 30.4769 | 32.6921 |
| BlogTe2 | 280 | **28** | **28** | 34 | **28** | 28.0234 | 22.3423 | 25.7373 | **22.0124** | 23.5863 |
| BlogTe3 | 280 | 39 | 50 | **28** | 55 | 93.7205 | 35.5453 | 34.6225 | **33.4716** | 35.1093 |
| BlogTe4 | 280 | 42 | **28** | 112 | 109 | 21.4296 | 10.3714 | **10.0159** | 11.2066 | 11.0762 |
| BlogTe5 | 280 | **28** | **28** | 78 | 40 | 22.1625 | 14.2714 | 15.2853 | **12.1534** | 13.5933 |
| BlogTe6 | 280 | 62 | 59 | 143 | **28** | 19.5266 | 16.9126 | **15.3457** | 19.2637 | 18.1872 |
| BlogTe7 | 280 | 53 | **28** | 39 | 151 | 11.9441 | 10.5470 | **10.3512** | 10.6481 | 11.9441 |
| BlogTe8 | 280 | **28** | 31 | 207 | 136 | 51.4217 | **36.9203** | 40.6439 | 51.42167 | 40.4043 |
| BlogTe9 | 280 | **28** | 50 | 31 | 61 | 35.0042 | 33.2960 | 32.5312 | **32.4372** | 33.1735 |
| BlogTe10 | 280 | 39 | **28** | 154 | 34 | 23.6604 | 21.9641 | **19.8671** | 23.3150 | 20.0649 |

**Note:**
Values in bold means best results.

## CMOPSO validity verification

### Validation of CMOPSO's validity on regular datasets

CMOPSO changes the acceleration factor and inertia weights basis on MOPSO, using dynamic acceleration factor and nonlinear decreasing inertia weights to make better diversity for particles to jump out of the local optimum in the early stage and speed up the particles in the later stage to improve the performance of MOPSO and make it have a better optimization finding accuracy. The regular dataset on UCI in Table 2 was used to verify the validity of CMOPSO and compared with two feature selection methods, approximate Markov blanket (AMB) and DA2MBL1, in the literature (*Li et al., 2022*). The DA2MBL1 method obtains similar feature groups by approximate Markov blanket clustering of features, imposes L1 regular term constraints on each similar feature group, and combines the coordinate descent method of solving to make the regression coefficients of the redundant features compressed to 0 to achieve the purpose of deleting redundant features. To avoid the chance of CMOPSO, each data is run ten times, and group's solution with the smallest $f_1$ or $f_2$ is taken each time. The average of ten times is finally taken, and the result after 10-fold cross-validation is shown in Table 7. Bolded text is the optimal value of the result.

Table 7, shows that the CMOSPO algorithm has better results on the SPmath, SPPort, and Rbuild datasets. Especially on the Rbuild dataset, y1 and y2 are left with only three features from 103 features after the CMOPSO algorithm removes irrelevant and redundant features, and the RMSE of y1 decreases from the original 589.1403 to 207.4673, and the RMSE of y2 decreases from the original 55.0247 to 34.8772. From the three datasets in Table 7, it can be seen that the CMOPSO algorithm has the lowest number of features and the lowest RMSE among the three algorithms, so CMOPSO may have better results in removing irrelevant and redundant features on regular datasets compared to AMB and DA2MBL1 algorithms.

**Table 7 Comparison of CMOPSO with other algorithms results obtained in regular datasets.**

| Dataset | Feature number (n) | | | | RMSE | | | |
|---|---|---|---|---|---|---|---|---|
| | Full set | AMB | DA2MBL1 | CMOPSO | Full set | AMB | DA2MBL1 | CMOPSO |
| SPmath-y1 | 30 | 18 | 18 | **10.5** | 3.1067 | 3.0444 | 2.9892 | **2.9604** |
| SPmath-y2 | 30 | 16 | 16 | **10.6** | 3.5856 | 3.5351 | 3.4435 | **3.4201** |
| SPmath-y3 | 30 | 13 | 13 | **10.2** | 4.4275 | 4.2591 | 4.2599 | **3.4214** |
| SPPort-y1 | 30 | 16 | 16 | **10.6** | 2.3530 | 2.3408 | 2.3033 | **2.2925** |
| SPPort-y2 | 30 | 16 | 16 | **10.7** | 2.5378 | 2.5197 | 2.4978 | **2.4797** |
| SPPort-y3 | 30 | 15 | 15 | **13.8** | 2.7920 | 2.7601 | 2.7533 | **2.7307** |
| Rbuild-y1 | 103 | 7 | 7 | **3** | 589.1403 | 243.2698 | 238.1848 | **207.4673** |
| Rbuild-y2 | 103 | 8 | 8 | **3** | 55.0247 | 45.3516 | 37.6467 | **34.8772** |

Note:
Values in bold means best results.

### Parameter setting

This article will involve two multi-objective optimization algorithms: MOPSO and Non-dominated Sorting Genetic Algorithm-II (NSGA-II) (*Deb et al., 2002*). NSGA-II is one of the popular multi-objective methods that uses a fast, non-dominated sorting algorithm and introduces an elitist strategy. To demonstrate the effectiveness of the MCMOPSO proposed in this article, the unmodified MOPSO and NSGA-II were used for experimental comparison. The number of iterations set for fairness is 300, the particle swarm size is 100, and the remaining parameters are set as shown in Table 8. The conventional parameters of MOPSO and CMOPSO are the same.

### Validation of MCMOPSO effectiveness on high dimensional small sample datasets

To address the feature selection problem for metabolomics high-dimensional small-sample data, this article proposes the MCMOPSO method using a two-stage feature selection framework, tested on two metabolomics data and 10 conventional high-dimensional small-sample data. In this section, two multi-objective methods, NSGA-II and MOPSO, are used to compare with the second-stage MCMOPSO. To better compare the three algorithms, all use mRMR to adaptively select the candidate feature set as input features and the number of feature subsets and RMSE as two optimization objectives. To avoid the chance of the algorithm results, each comparison algorithm is run independently ten times on each dataset. The best Pareto solution with the same number of features in the results of the ten runs is averaged, and the smaller the RMSE is, the better it is for the same number of features. The final results are presented in a bar chart, and the error line is added to indicate the difference between the results of the ten runs and the average. This time, the error metric in the graph is standard deviation, and the shorter length of the error line indicates that the results of each run are very close to the mean and the algorithm has good stability.

Figure 4 shows four endogenous substance data to verify the validity of MCMOPSO. The horizontal coordinate of this bar graph indicates the number of feature subsets, and

**Table 8 Information on the parameterization of the two algorithms.**

| Algorithm | Parameters |
| --- | --- |
| CMOPSO | T = 300, nPop = 100, Wmax = 0.9, Wmin = 0.4, Mesh = 20, nAr = 50 |
| NSGA-II | T = 300, nPop = 100, crossover probability = 0.6, mutation probability = 0.1 |

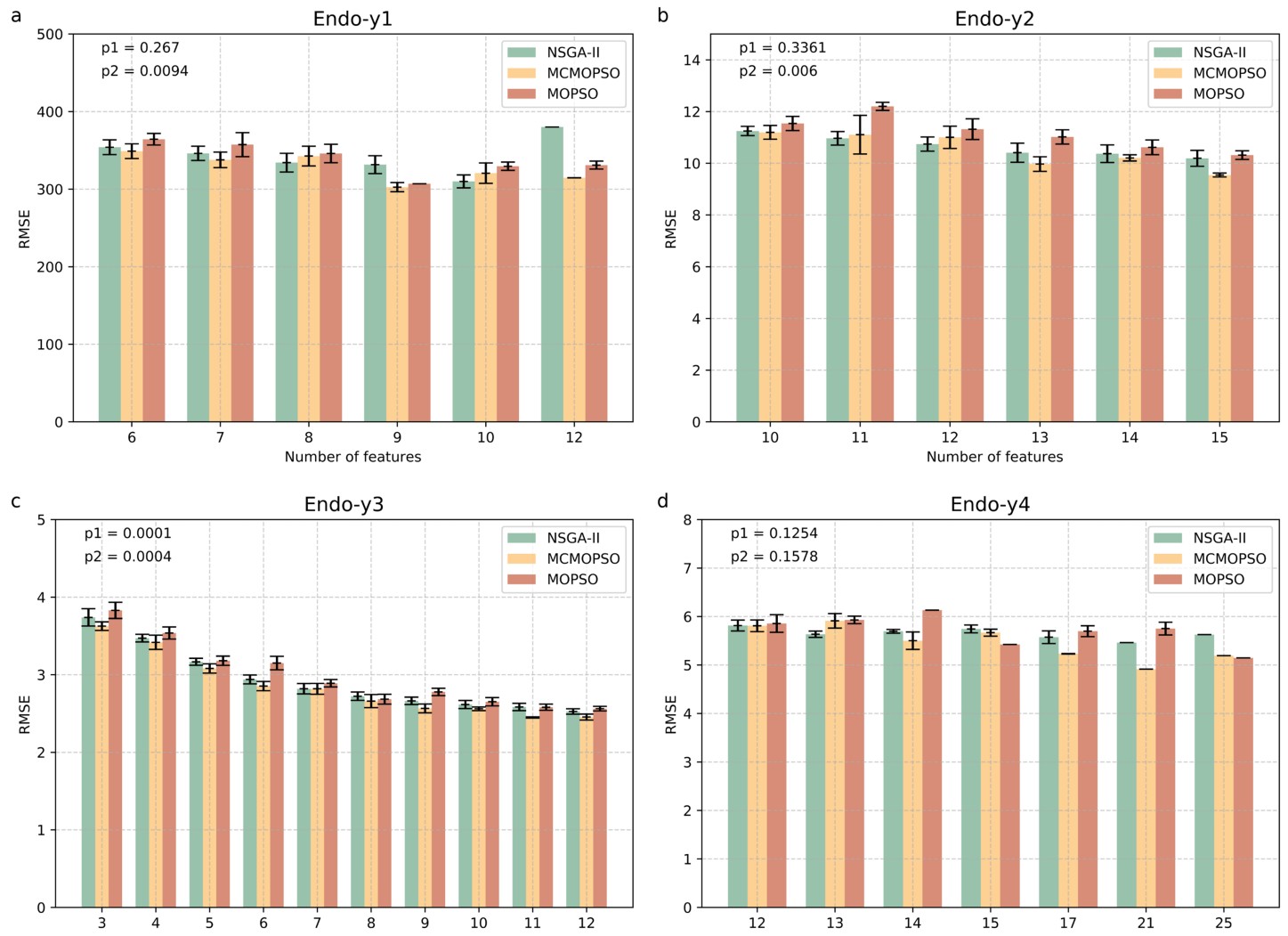

**Figure 4 Comparison of experimental results of MCMOPSO, NSGA-II and MOPSO algorithms on endogenous substances.** Where the horizontal coordinate number of features is the number of output features, the vertical coordinate is the root mean square error (RMSE) of the regression's evaluation metrics, and the p1 in the upper right corner is the *p*-value obtained from hypothesis testing of the algorithms MCMOPSO and NSGA-II is denoted as p1, and the *p*-value obtained from hypothesis testing of the MCMOPSO algorithm and the MOPSO algorithm is denoted as p2.               

the vertical coordinate indicates the RMSE size. As the three algorithms perform better when the number of features in the Endo-y1 data is taken as 9 and 12, the average RMSE of NSGA-II is 331.53 and 380.02, the average RMSE of MCMOPSO is 302.51 and 314.65,

respectively, and the average RMSE of MOPSO is 306.85 and 330.95, respectively. For NSGA-II, MCMOPSO and MOPSO have error values of 23.085, 11.78, and 0 for 9 features and 0, 0, and 10.34 for 12 features, respectively. To compare of the mean RMSE, the MCMOPSO algorithm performs the best, and the NSGA-II algorithm performs the worst for both features. As for the error line, each algorithm's performance varies for different features. CMOPSO algorithm has a low error line for nine features, and the MOPSO algorithm has a low error line for 12 features. However, the NSGA-II algorithm has a higher error line for both features.

As can be observed from the results in Fig. 4, the average RMSE of MCMOPSO on the four endogenous substances data is smaller. Still, when Endo-y1 retains eight ions, the RMSE is higher than that of NSGA-II but slightly lower than that of MOPSO with a similar magnitude of error. In the Endo-y1 data, MCMOPSO's average RMSE is better than the two comparison algorithms, but the error is more significant when 11 ions are selected. MCMOPSO in the Endo-y3 data outperforms the comparison algorithms in the average RMSE and the error value. The average RMSE is lower than NSGA-II on the Endo-y4 data except for selecting 13 ions, which is lower than the average RMSE of NSGA-II and MOPSO.

In addition, in this article, we conducted independent t-test for the MCMOPSO algorithm against the NSGA-II algorithm and the MOPSO algorithm, respectively, and obtained the corresponding $p$-values, denoted as p1 and p2, respectively, which are shown in the upper right corner of each figure in Fig. 4. In addition to performing hypothesis tests, we obtained two other evaluation metrics through regression analysis: the R-square and the MAE. We averaged the solutions obtained from one run and ten runs for each dataset and computed the final average of their R-squared and MAE. The results for R-squared and MAE are tabulated for comparison in File S1 (Table S5). From the $p$-value results for datasets y1, y2, and y3, our algorithm shows a significant advantage ($p$-value < 0.05) under certain comparison conditions. While the R-square for all four datasets is larger than the two comparison algorithms, the MAE for y1, y3, and y4 is smaller than the two.

In summary, the MCMOPSO algorithm outperforms the two comparison algorithms on endogenous substances overall, but the significance difference is less pronounced and only significant on some of the datasets.

Figure 5 shows the results of the runs on the four pharmacodynamic indicators in exogenous substances; from the figure MCMOPSO algorithm on Exo-y1, Exo-y3, and Exo-y4, it can be seen that the average RMSE is lower than the two comparative algorithms and better performance. Still, on Exo-y2, the performance is not as good as the NSGA-II algorithm but better than the MOPSO algorithm. From the error line, it can be seen that MCMOPSO has fewer errors on Exo-y2-y4 but more on Exo-y1, especially when 10 ions are selected. The MCMOPSO algorithm outperforms NSGA-II and MOPSO algorithms on exogenous substance data.

From the $p$-values obtained during the hypothesis testing, all $p$-values are less than 0.1 except for p2 on y1, which is greater than 0.1. and p1 on the four datasets of exogenous substances are less than 0.05. From the results of R-squared and MAE, the R-squared of y2, y3, and y4 are more significant than the two comparative algorithms, and the MAE of y2

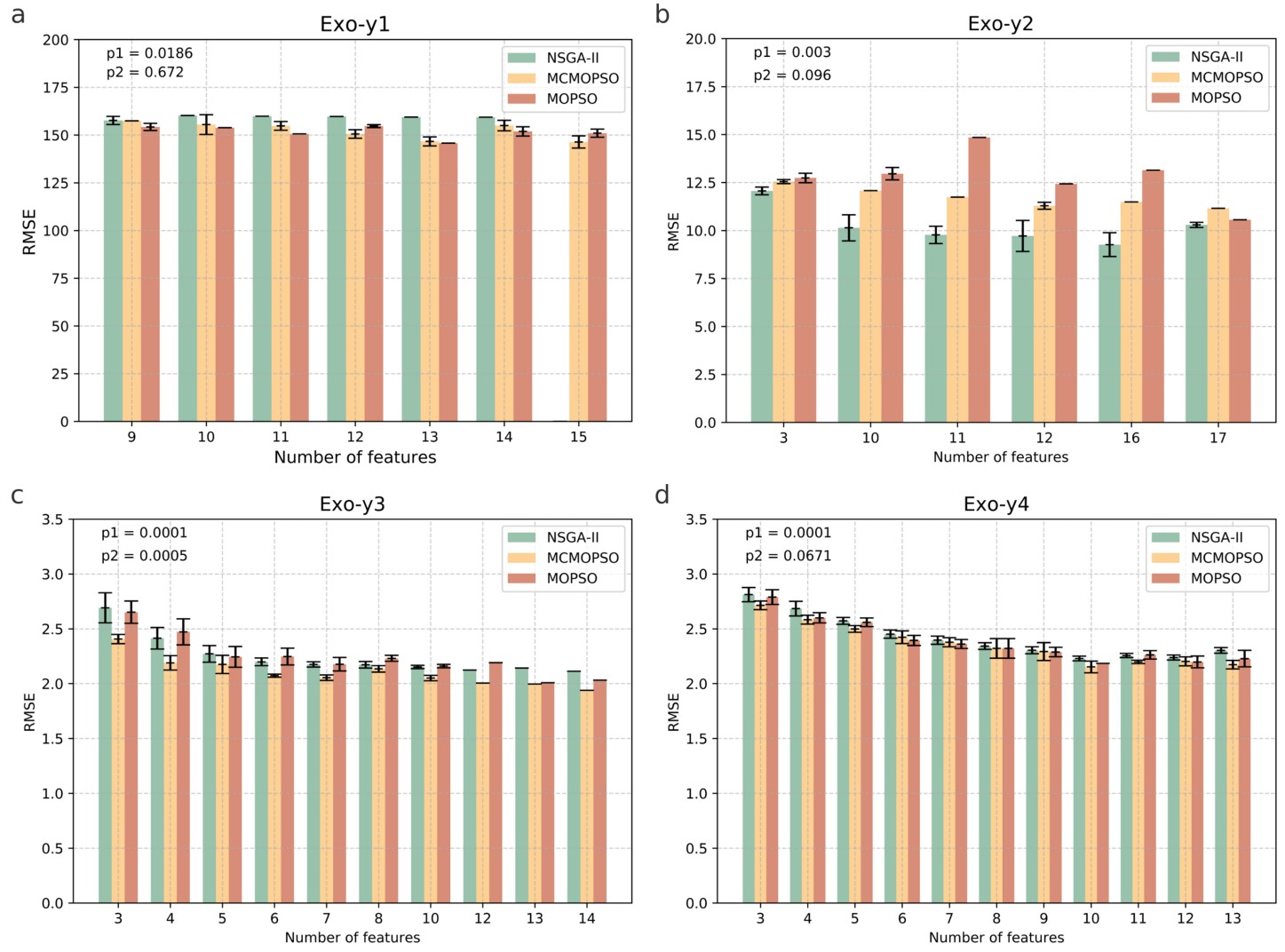

**Figure 5 Comparison of experimental results of MCMOPSO, NSGA-II and MOPSO algorithms on exogenous substances.** The p1 in the upper right corner is the *p*-value obtained from hypothesis testing of the algorithms MCMOPSO and NSGA-II is denoted as p1, and the *p*-value obtained from hypothesis testing of the MCMOPSO algorithm and the MOPSO algorithm is denoted as p2.

and y3 are less than the two comparative algorithms. There is a substantial difference between MCMOPSO and NSGA-II in the exogenous substance dataset, but the critical difference with MOPSO on this dataset is not so noticeable. From the results of multiple regression evaluation metrics, it can be seen that MCMOPSO outperforms the two comparative algorithms in most of the datasets and, therefore, outperforms them in overall performance.

Figure 6 shows the experimental results on ten regular HDSS datasets. The MCMOPSO algorithm significantly outperforms the NSGA-II and MOPSO algorithms regarding average RMSE and error on data BlogTe1-3, BlogTe6, and BlogTe10. The BloTe5 algorithm does not perform as well as the two comparative algorithms when taking three

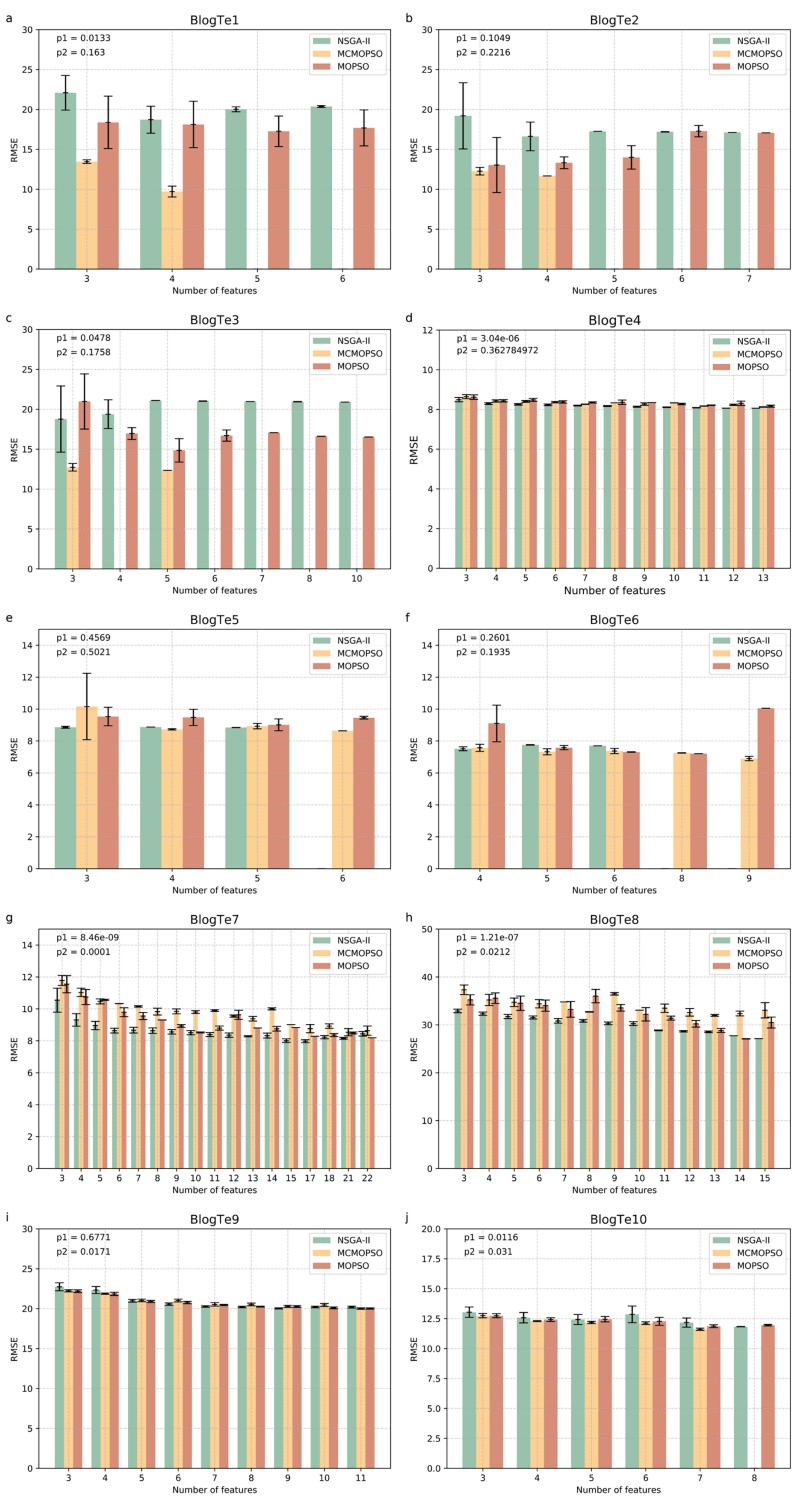

**Figure 6 Comparison of experimental results of MCMOPSO, NSGA-II and MOPSO algorithms on UCI substances.** The figure depicts a comparison of the results of MCMOPSO and the two comparison algorithms on regular high-dimensional small-sample data in UCI. The p1 in the upper right corner is the *p*-value obtained from hypothesis testing of the algorithms MCMOPSO and NSGA-II is denoted as p1, and the *p*-value obtained from hypothesis testing of the MCMOPSO algorithm and the MOPSO algorithm is denoted as p2.

ions but outperforms them when taking 4–6 ions. Similarly, on BlogTe9 data the MCMOPSO algorithm has a lower average RMSE than NSGA-II and MOPSO algorithms when the number of ions taken is three, four, and 11. On BlogTe4, BlogTe7, and BlogTe8, the MCMOPSO algorithm does not perform as well as the two comparison algorithms, but the errors are similar.

The p1 value is less than 0.05 for six data on the public dataset, *i.e.*, more than half of the datasets have a significant difference between MCMOPSO and NSGA-II, and the p2 value is less than 0.05 for four datasets. From the results of the R-square and the MAE, it can be seen that the R-square of the MCMOPSO algorithm outperforms the two comparative algorithms for five datasets, and the MAE outperforms the two comparative algorithms for six datasets (Table S6 in File S1). Therefore, the MCMOPSO algorithm is slightly better than the NSGA-II algorithm and MOPSO algorithm in overall performance.

In summary, the MCMOPSO algorithm proposed in this article is slightly better than the NSGA-II and MOPSO algorithms in dealing with HDSS data, and the error is more minor in most cases. However, there is a significant error in some data or when choosing a specific number of features, so the algorithm's stability needs to be improved. The experimental comparison shows that the difference between NSGA-II, MCMOPSO, and MOPSO is more evident in the Exo dataset, especially the BlogTe dataset. This is because the Endo dataset has higher dimensionality than the Exo and BlogTe datasets, and NSGA-II maintains the diversity of the population by using non-dominated ordering and crowding degree distance. More dimensions provide more possibilities and diversity in higher dimensional space due to larger solution space. So, NSGAII works slightly better with higher dimensional endo datasets. Different algorithms may show different advantages and disadvantages for a MOOP like feature selection. This leads to more apparent differences between NSGA-II, CMOPSP, and MOPSO on Exo datasets, especially BlogTe datasets.

### Comparison of algorithm performance

Time efficiency is also an essential element of feature selection research, and this article will compare and analyze the algorithm's running time and time complexity. Table 9 lists the running times of the proposed algorithm and the two comparison algorithms. The table shows that CMOPSO has much less running time than NSGA-II on regular high-dimensional data and higher-dimensional data in metabolomics, with 13 of them having less running time than MOPSO. Because CMOPSO changes the acceleration factor based on the original MOPSO, it is a little faster and takes less time to run. In terms of time complexity, assuming that the size of the particles in the particle swarm and the size of the population in NSGA-II are both n, and the particle size and initial population size are m, the time complexity of MOPSO and CMOPSO is $\max[O(mn), O(n), O(n^2)]$, and the time complexity of NSGA-II is $\max[O(mn), O(n^2), O(n^3)]$. Comparing the time complexity, the time complexity of MOPSO is lower than that of NSGA-II. The experimental results show that CMOPSO has better calculation efficiency in the feature selection algorithm based on multi-objective optimization.

**Table 9  Running time of the three algorithms in the second stage.**

| DataSet | Algorithm | | |
| --- | --- | --- | --- |
| | MCMOPSO (s) | MOPSO (s) | NSGA-II (s) |
| Endo-y1 | **311.9949** | 348.8238 | 441.95 |
| Endo-y2 | **315.7805** | 380.2408 | 421.7625 |
| Endo-y3 | **301.7312** | 364.3446 | 421.4107 |
| Endo-y4 | **336.0243** | 354.5867 | 550.6532 |
| Exo-y1 | 359.8688 | **331.4467** | 452.0622 |
| Exo-y2 | **396.169** | 412.7382 | 607.35 |
| Exo-y3 | **334.2242** | 357.2573 | 437.8177 |
| Exo-y4 | 357.6967 | **337.1848** | 428.6405 |
| BlogTe1 | **228.7358** | 262.2551 | 409.0935 |
| BlogTe2 | 273.6847 | **241.3397** | 726.1251 |
| BlogTe3 | 300.2104 | **262.8484** | 726.9779 |
| BlogTe4 | 302.9208 | **280.6899** | 478.1777 |
| BlogTe5 | **267.2119** | 317.7942 | 456.3005 |
| BlogTe6 | **263.7787** | 312.7636 | 443.9855 |
| BlogTe7 | **325.7074** | 414.6876 | 546.4352 |
| BlogTe8 | **354.4535** | 407.3553 | 559.4451 |
| BlogTe9 | **374.816** | 378.5011 | 544.0629 |
| BlogTe10 | **284.06** | 342.6711 | 457.52 |

**Note:**
Values in bold means best results.

To present a more intuitive comparison of the running time of the three algorithms, the time comparison in Table 9 is visualized in the form of a bar chart, and the results are shown in Fig. 7. The horizontal coordinates represent different datasets, the vertical coordinates represent the running time, and the three colors represent the three algorithms. From the figure, it can be seen that the running time of the NSGA-II algorithm is greater than that of CMOPSO and MOPSO algorithms, whereas, on the datasets Endo-y1-y4, Exo-y2-y3, BlogTe1, and BlogTe5-10, the running time of CMOPSO algorithm is less than that of MOPSO algorithm. Therefore, the CMOPSO algorithm is slightly better than the two compared algorithms in terms of time performance.

## Comparison of MCMOPSO and other two-stage algorithms

MCMOPSO is a two-stage hybrid feature selection algorithm that integrates the Filter and Wrapper techniques. The core functionality of MCMOPSO is implemented in the second stage, Wrapper. Therefore, for experimental completeness, MCMOPSO was compared with two other two-stage feature selection algorithms, FCBF (*Li, Yu-Yu & Cong, 2018*) and CI_AMB (*Huang et al., 2020*). FCBF is an exemplary algorithm that operates in two distinct stages. In the first stage, this method utilizes symmetric uncertainty to compute the correlation between features and the target feature. Remove irrelevant features from the feature subset by setting a threshold value. During the second phase, FCBF employs the approximate Markov blanket technique to eliminate redundant features. CI_AMB is a

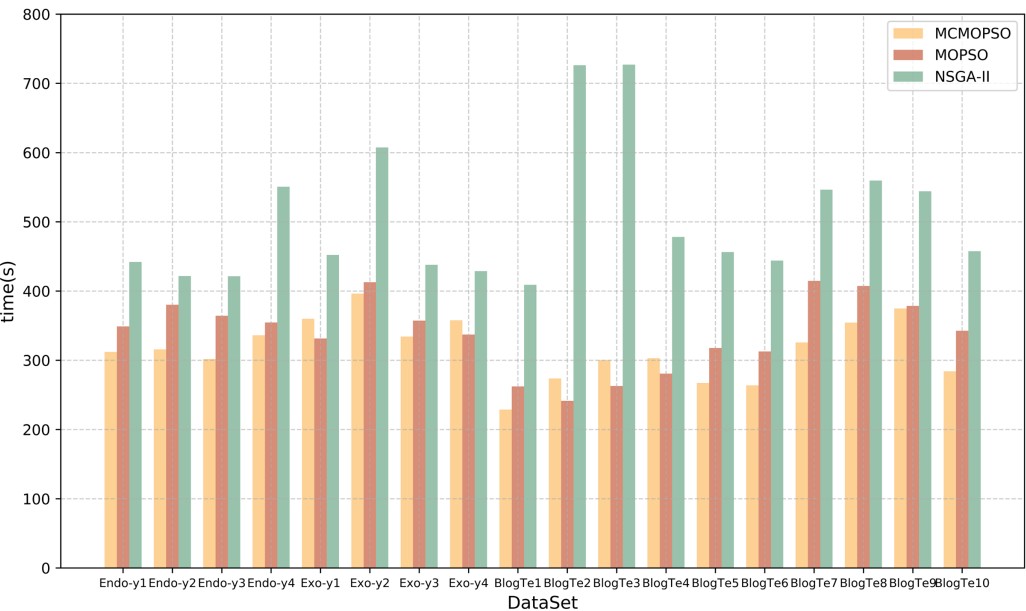

**Figure 7 Runtime comparison of MCMOPSO with MOPSO and NSGA-II algorithms.** The chart depicts the running times of the three compared algorithms.

hybrid feature selection method based on an iterative approximation of the Markov blanket. This method begins by measuring the correlation between features and the target variable using the MIC. It then removes irrelevant features based on evaluation criteria before employing the iterative approximation of the Markov blanket strategy to eliminate redundant features. Next, the Markov blanket strategy utilizes an iterative approach to gradually remove redundant features.

To better compare with the two-stage algorithms, the number of ions with low average RMSE is selected in Figs. 4 and 5 to compare the number of ions screened and the RMSE obtained by the CI_AMB and FCBF algorithms. In BlogTeFeedback data, MCMOPSO chooses the group of solutions with fewer features to compare with the two comparison algorithms, and the results are shown in Table 10; the bolded values are the optimal values.

Table 10, shows that among the eight metabolomics data, MCMOPSO chooses fewer ions than CI_AMB and FCBF, and the model accuracy is higher than theirs. For example, on the Endo-y1 data, MCMOPSO finally selects nine ions with an RMSE of 302.5077, while CI_AMB selects 64 ions with an RMSE of 705.3395. It is better than FCBF but not as good as MCMOPSO. In the BlogTeFeedback data of the UCI dataset, BlogTe1, BlogTe3, and BlogTe3, and BlogTe3, and BlogTe3 and BlogTe3, respectively, have a higher accuracy than FCBF, BlogTe3 and BlogTe6 on both objectives outperform the two compared algorithms, and on BlogTe2, BlogTe4-5 and BlogTe7-10, the model accuracy of MCMOPSO is higher under the condition of the same number of filtered features. In summary, MCMOPSO outperforms both two-stage algorithms with better dimensionality reduction and improved model accuracy.

**Table 10 Comparison of results between MCMOPSO and other two-stage algorithms on high-dimensional small sample datasets.** Comparison results of the two-stage algorithm MCMOPSO with more other algorithms.

| Dataset | Feature number (n) | | | | RMSE | | | |
|---|---|---|---|---|---|---|---|---|
| | Full set | CI_AMB | FCBF | MCMOPSO | Full set | CI_AMB | FCBF | MCMOPSO |
| Endo-y1 | 10,283 | 64 | 71 | **9** | 741.7310 | 705.3395 | 753.7210 | **302.5077** |
| Endo-y2 | 10,283 | 47 | 55 | **13** | 29.4566 | 24.3540 | 24.7824 | **9.9684** |
| Endo-y3 | 10,283 | 44 | 51 | **11** | 5.1288 | 5.1372 | 5.3590 | **2.4480** |
| Endo-y4 | 10,283 | 42 | 49 | **21** | 10.9573 | 10.9771 | 11.7237 | **4.9154** |
| Exo-y1 | 798 | 78 | 78 | **15** | 352.0239 | 288.3768 | 288.3768 | **146.3726** |
| Exo-y2 | 798 | 8 | 15 | **17** | 19.0489 | 15.7140 | 17.0696 | **11.1565** |
| Exo-y3 | 798 | **5** | 18 | 14 | 3.5128 | 3.4236 | 4.5020 | **1.9396** |
| Exo-y4 | 798 | 13 | 16 | **10** | 3.6348 | 3.7231 | 3.6980 | **2.1526** |
| BlogTe1 | 280 | 5 | 5 | **4** | 36.3218 | 28.1551 | 37.7140 | **9.7105** |
| BlogTe2 | 280 | **4** | 4 | 4 | 28.0234 | 19.4269 | 33.2647 | **11.6698** |
| BlogTe3 | 280 | 4 | 5 | **3** | 93.7205 | 30.6052 | 42.0285 | **12.7244** |
| BlogTe4 | 280 | **3** | 5 | 3 | 21.4296 | 9.9083 | 11.2857 | **8.6421** |
| BlogTe5 | 280 | 4 | 4 | **4** | 22.1625 | 11.4452 | 11.4451 | **8.7269** |
| BlogTe6 | 280 | 7 | 7 | **6** | 19.5266 | 18.8514 | 18.8514 | **7.3704** |
| BlogTe7 | 280 | 7 | 7 | 7 | 11.9441 | 10.1605 | 12.1941 | **10.1536** |
| BlogTe8 | 280 | **4** | 4 | 4 | 51.4217 | 41.5088 | 41.9495 | **35.1875** |
| BlogTe9 | 280 | **4** | 4 | 4 | 35.0042 | 28.2894 | 33.5649 | **20.2559** |
| BlogTe10 | 280 | **5** | 5 | 5 | 23.6604 | 18.8846 | 19.2383 | **12.1774** |

**Note:**
Values in bold means best results.

In addition, this article was compared with the regression metabolomics data from the literature (*Acharjee et al., 2020*), which compared several methods, and in the end, Boruta's method gave the best results. Boruta's method selected 46 features from metabolomics and reduced them to seven after power tools. The MCMOPSO algorithm uses the same dataset. In the initial stage, the mRMR algorithm selects 19 features, while in the second stage, the CMOPSO algorithm obtains a set of solutions where the minimum number of features is 3. To compare the performance of the two models, MCMOPSO adopts the same way of evaluating the model performance as in the comparison article, and the regressor uses random forests and keeps the same parameters. The final R-square of the three features selected by MCMOPSO was 0.8624, while Boruta's method yielded an R-square of 0.842. In other words, we have chosen fewer features than mentioned in the literature with higher R-squared values.

## DISCUSSION

A series of valuable conclusions are drawn by comparing the performance of three multi-objective feature selection methods, MCMOPSO, MOPSO, and NSGAII, on HDSS data for metabolomics. Firstly, it can be observed that the MCMOPSO method effectively filters a condensed subset of features on most datasets, demonstrating its superiority in the feature selection task. The MCMOPSO method also offers significant advantages regarding the

quality of feature subsets compared to traditional MOPSO and NSGAII. By evaluating the selected feature subset in the regression task using two regression evaluation metrics, R-squared and MAE, it is found that MCMOPSO performs better on most datasets. This further validates the effectiveness of the MCMOPSO method. A high-quality feature subset not only helps improve the regressor's performance but also reduces the risk of model overfitting, which improves the model's generalization ability. In addition, the study reveals the time-efficiency advantage of the MCMOPSO method. By adopting a series of optimization strategies and heuristic algorithms, MCMOPSO can significantly reduce the computation time during the feature selection process, making it more feasible and practical in practical applications. This efficient feature selection method support for metabolomics research and other HDSS analysis.

Many similar studies have applied feature selection to the field of metabolomics. For example, *Prete et al. (2016)* used a feature selection approach to analyze protein features to determine their relationships and specificity within protein families. *Chardin et al. (2021)* proposed a new feature selection classification method (PD-CR) to analyze two metabolomics data: urine and samples of mutant isocitrate dehydrogenase (IDH) or wild-type IDH from lung cancer patients and healthy controls. It was also compared with PLS-DA, Random Forest, and SVM algorithms. The results show that the advantage of PD-CR is that it provides a confidence score for each prediction that can be used for culling classification. This significantly reduces the false discovery rate. *Grissa et al. (2016)* utilized knowledge discovery and data mining methods to propose advanced solutions for predictive biomarker discovery. The strategy evaluates combined numerical and symbolic feature selection methods to obtain the optimal combination of metabolites that produce effective and accurate predictive models. *Fu et al. (2020)* performed feature selection and classification for class-imbalanced data in metabolomics by minimizing the degree of overlap. The results show that the proposed algorithm effectively identifies key features and controls false discovery for class balance learning.

Despite the strengths of our study, we are aware of its limitations. For example, our experiments may be limited by the size and quality of the dataset, which may cause our results to be somewhat biased. Future research directions will focus on further improving our method. For example, more complex optimization algorithms or combining multiple algorithms could be explored to enhance the performance of feature selection. In addition, we plan to incorporate other data information in metabolomics, such as mz values and retention times, to confirm biomarkers more comprehensively. By combining different types of data information, the accuracy and reliability of biomarkers can be improved for better application in metabolomics research.

Our study provides an effective method for solving the feature selection problem in analyzing HDSS data in metabolomics. Although there are still some limitations, through future improvements and extensions, we believe this method will show a broader development prospect in future applications and make a more significant contribution to the progress of metabolomics research and the biomedical field.

## CONCLUSION

To address the issue of the MOPSO algorithm being prone to get trapped in local optima, this article proposes the utilization of dynamic acceleration factors and nonlinearly decreasing inertia weights. These measures aim to help the algorithm escape local optima and enhance its convergence capabilities. Facing the problems caused by high dimensional small sample data containing more noisy data and too high dimensionality, a hybrid mRMR and multi-objective particle swarm feature selection algorithm (MCMOPSO) is proposed. In the initial phase, a Filter-Wrapper hybrid approach is employed to remove irrelevant and partially redundant features dynamically. The second phase, the CMOPSO algorithm developed in this study, is utilized to eradicate the remaining redundant features further. It has been established through comprehensive experimental design and comparative analysis that MCMOPSO operates with remarkably efficiently. It effectively eliminates irrelevant features and minimizing redundant ones, resulting in identifying a highly refined subset of features. MCMOPSO has been well used in both UCI datasets and metabolomics data. Moreover, it performs even better when MCMOPSO faces metabolomics data with higher feature dimensions and smaller sample sizes. It can effectively screen out a small number of ions with high quality, providing technical support for data processing in metabolomics. Moving forward, our focus will be on further improving the algorithm to enhance the stability of the model while reducing the time consumption and continuing to search for better multi-objective feature selection methods.

## ABBREVIATIONS

| | |
|---|---|
| **mRMR** | Max-Relevance and Min-Redundancy |
| **MCMOPSO** | hybrid Max-Relevance and Min-Redundancy (mRMR) and multi-objective particle swarm feature selection method |
| **CMOPSO** | multi-objective particle swarm algorithm based on dynamic linear adjustment of acceleration factors and nonlinear decreasing weight coefficients |
| **PLS** | partial least squares |
| **MOPSO** | multi-objective particle swarm |
| **NSGA-II** | Non-dominated Sorting Genetic Algorithm-II |
| **RMSE** | root mean square error |
| **HDSS** | high-dimensional small samples |
| **MOOP** | multi-objective optimization problem |
| **MAE** | mean absolute error. |

### Funding

This work was supported by the National Natural Science Foundation of China (No. 82260988, No. 82274680 and No. 82160955) and the Jiangxi University of Chinese

Medicine Science and Technology Innovation Team Development Program (Grant No CXTD22015). The funders had no role in study design, data collection and analysis, decision to publish, or preparation of the manuscript.

### Grant Disclosures
The following grant information was disclosed by the authors:
National Natural Science Foundation of China: 82260988, 82274680 and 82160955.
Jiangxi University of Chinese Medicine Science and Technology Innovation Team Development Program: CXTD22015.

### Competing Interests
The authors declare that they have no competing interests.

### Author Contributions
- Mengting Zhang conceived and designed the experiments, performed the experiments, analyzed the data, performed the computation work, prepared figures and/or tables, and approved the final draft.
- Jianqiang Du conceived and designed the experiments, authored or reviewed drafts of the article, and approved the final draft.
- Bin Nie performed the experiments, authored or reviewed drafts of the article, and approved the final draft.
- Jigen Luo performed the experiments, authored or reviewed drafts of the article, and approved the final draft.
- Ming Liu analyzed the data, prepared figures and/or tables, and approved the final draft.
- Yang Yuan performed the computation work, prepared figures and/or tables, and approved the final draft.

### Data Availability
The code is available at GitHub and Zenodo:
- https://github.com/zmtCode/MCMOPSO.git.
- zmtCode. (2024). zmtCode/MCMOPSO: v1.0.0 (v1.0.0). Zenodo. https://doi.org/10.5281/zenodo.11115056.

### Supplemental Information
Supplemental information for this article can be found online at http://dx.doi.org/10.7717/peerj-cs.2073#supplemental-information.

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
