# Peer review of "Hybrid mRMR and multi-objective particle swarm feature selection methods and application to metabolomics of traditional Chinese medicine"

_PeerJ Computer Science, doi:10.7717/peerj-cs.2073_

## Round 0.1 · original submission · Major Revisions

The reviewers have substantial concerns about this manuscript. The authors should provide point-to-point responses to address all the concerns and provide a revised manuscript with the revised parts being marked in different color.

**Language Note:** The review process has identified that the English language must be improved. PeerJ can provide language editing services - please contact us at [email protected] for pricing (be sure to provide your manuscript number and title). Alternatively, you should make your own arrangements to improve the language quality and provide details in your response letter. – PeerJ Staff

Reviewer 1 ·

Basic reporting

Zhang et al proposed a new method named MCMOPSO that used for metabolomics data processing. Metabolimics is an important component of systems biology. Within the framework of genemoe, transcriptome, protomes, metabolomics, metabolomics is at the most downstream and only metabolomics can truly reflects what had happened. However, it is difficult to dig out the true target from metabolimics dataset as it is a typical HDSS data. Overall, the manuscript is well written and organized, and the proposed method is applicable for related research. However, I have several minor concerns need to be addressed by the authors.

Experimental design

The presented manuscript fit the aim and scope of peerj.

Validity of the findings

1. I am a little bit about the metabolimics data from the ginseng injection. Which figure is it related to? Also, if this is a new dataset, the detailed data need to be supplemented, such as the LC-MS-QTOF related data.
2. More significant difference between NSGA-II, CMOPSP and MOPSO was found in exo and especially blogte dataset. Authors should explain what lead to the difference, such as the different sensitivity of different method in different dataset or other reasons.

Additional comments

Some figures need to be prepared in a better way. For example, Exo-y3/Exo-y4 is too crowded in fig 5. Same was found for BlogTe4, 7, 8 in fig 6. It would be better for publication if authors can redraw fig 4-fig 7 in other way such as R, instead of excel (they look like is generated from excel).

Reviewer 2 ·

Basic reporting

The study is well-designed and straightforward. However, I have several suggestions for improvement.
I strongly recommend that the authors refer to the author instructions provided by PeerJ. Using the PeerJ template for research papers could enhance the organization and presentation of your manuscript. A reorganization of the current sections is necessary.

The manuscript would benefit from a thorough review to improve clarity and coherence. Authors should ensure that their arguments are logically structured and clearly articulated. Certain sections require language refinement to achieve a more academic tone and enhance precision. Revisiting these sections could significantly improve the quality of writing. The text includes ambiguous phrases that could lead to confusion. It would be beneficial for the authors to revise these phrases for clearer communication of the intended messages. Complex and convoluted sentences should be simplified to enhance the readability and accessibility of the paper. The manuscript appears to require proofreading to address grammatical errors and typos.

Experimental design

Could the authors elaborate in the data section on their rationale for selecting those three conventional datasets to assess the validity of the CMOPSO? These datasets appear markedly different from those typically used in metabolomics studies.

Validity of the findings

In terms of figures and tables, please ensure that "Stage 1" and "Stage 2" are clearly labeled in Figure 1. Additionally, the rows and columns in Tables 5, 7, and 10 should be aligned for clarity and consistency.

All underlying data have been provided; they are robust, statistically sound, & controlled. Conclusions are well stated, linked to original research question & limited to supporting results.

I highly recommend authors write a discussion section to include information provided in the current relevant work session.

Additional comments

I have provided detailed comments for highlighted content.

Annotated reviews are not available for download in order to protect the identity of reviewers who chose to remain anonymous.

Reviewer 3 ·

Basic reporting

The paper is written in clear, unambiguous, professional English language, making it accessible to readers. The introduction provides a comprehensive background, situating the research within the context of current knowledge and demonstrating the relevance of the study. The literature is well-referenced and pertinent to the research question, which aims to address challenges in metabolomics data analysis. The structure of the paper conforms to standards, with high-quality figures and tables that are well-labeled and described.

Experimental design

The research presents original primary research within the scope of the journal. The research question is well-defined, relevant, and meaningful, clearly stating how the research fills an identified knowledge gap. A rigorous investigation has been performed to a high technical and ethical standard. Methods are described with sufficient detail to allow replication, meeting the requirements for experimental design.

Validity of the findings

The findings demonstrate the effectiveness of the proposed hybrid feature selection method (MCMOPSO) in handling high-dimensional small sample (HDSS) data in metabolomics.

Additional comments

The paper is well-organized and logically structured, providing a significant contribution to the field of feature selection in metabolomics. The proposed MCMOPSO method is a novel approach that effectively addresses the challenges of HDSS data, demonstrating improved performance over existing methods. The extensive experimental analysis using various datasets supports the validity of the proposed method. However, it would be beneficial if future work could explore the applicability of MCMOPSO to other types of biological data beyond metabolomics.

---

## Round 0.2 · accepted · Accept

Reviewers are satisfied with the revisions, and I concur to recommend accepting this manuscript.

Reviewer 1 ·

Basic reporting

All my comments suggestions were addressed in full.

Experimental design

The experiments are well designed.

Validity of the findings

The finding is useful to the field.

Additional comments

All my questions have been addressed. I don't have any more questions. The manuscript is suitable for publication.

Reviewer 2 ·

Basic reporting

no comment

Experimental design

no comment

Validity of the findings

no comment